# Identification of Regions Involved in the Physical Interaction between Melanocortin Receptor Accessory Protein 2 and Prokineticin Receptor 2

**DOI:** 10.3390/biom12030474

**Published:** 2022-03-20

**Authors:** Maria Rosaria Fullone, Daniela Maftei, Martina Vincenzi, Roberta Lattanzi, Rossella Miele

**Affiliations:** 1Department of Biochemical Sciences “Alessandro Rossi Fanelli”, Sapienza University of Rome, Piazzale Aldo Moro 5, 00185 Rome, Italy; mariarosaria.fullone@uniroma1.it; 2Department of Physiology and Pharmacology “Vittorio Erspamer”, Sapienza University of Rome, Piazzale Aldo Moro 5, 00185 Rome, Italy; daniela.maftei@uniroma1.it (D.M.); martina.vincenzi@uniroma1.it (M.V.); roberta.lattanzi@uniroma1.it (R.L.)

**Keywords:** prokineticins, MRAP2, obesity, GPCR, STAT3, ERK

## Abstract

Melanocortin Receptor Accessory Protein 2 (MRAP2) modulates the trafficking and signal transduction of several G-protein-coupled receptors (GPCRs) involved in the control of energy homeostasis, such as Prokineticin receptors (PKRs). They bind the endogenous ligand prokineticin 2 (PK2), a novel adipokine that has an anorexic effect and modulates thermoregulation and energy homeostasis. In the present work, we used biochemical techniques to analyze the mechanism of interaction of MRAP2 with PKR2 and we identified the specific amino acid regions involved in the complex formation. Our results indicate that MRAP2 likely binds to the N-terminal region of PKR2, preventing glycosylation and consequently the correct receptor localization. We also identified a C-terminal region of MRAP2 that is critical for the interaction with PKR2. Consequently, we analyzed the role of the prokineticin transduction system in the regulation of MRAP2 expression in tissues involved in the control of food intake: at the central level, in hypothalamic explants, and at the peripheral level, in adipocytes. We demonstrated the modulation of MRAP2 expression by the prokineticin transduction system.

## 1. Introduction

The Melanocortin Receptor Accessory Protein (MRAP) family, which includes two members, MRAP1 and MRAP2, binds G protein-coupled receptors (GPCRs) and modulates their trafficking and signaling transduction. MRAP2 is a small protein with a single hydrophobic transmembrane domain that can dimerize in parallel and antiparallel orientations. This dual topology is a unique feature in the eukaryote proteome [1].

MRAP2 knock-out (KO) mice develop severe early onset obesity because MRAP2 is involved in the control of food intake [2]. Namely, it regulates several GPCRs, such as the melanocortin-4 receptor (MC4R), orexin receptor 1, and ghrelin receptor, which are critical for the control of energy homeostasis [3]. 

MRAP2 also regulates prokineticin receptors (PKRs) by reducing their signaling pathway and plasmatic membrane localization [4]. It was possible to distinguish two distinct parts of the C-terminal region of MRAP2 for the modulation of prokineticin receptors; the region from 76 to 141 is important for both trafficking and signaling, whereas the region from amino acid 141 to 205 is important only for signaling [5].

PKRs belong to GPCRs that are expressed both centrally and peripherally [6]. PKR2 is mainly localized in the central nervous system (CNS), whereas PKR1 is found in peripheral tissues [6]. PKRs couple to G_αq/11_, G_αs_ and G_αi_ after prokineticin 2 (PK2) binding [7], and mediate several signaling pathways that promote increases in intracellular calcium and cAMP levels, Akt phosphorylation, and the activation of ERK and STAT3 [7,8]. 

In neurons of the hypothalamic arcuate nucleus, PK2 induces STAT3 activation and melanocyte-stimulating hormone release by binding to PKR2, resulting in an anorectic effect and modulating thermoregulation and energy homeostasis. In adipocytes, PK2, which binds to PKR1, limits preadipocyte proliferation and differentiation, thereby controlling adipose tissue expansion [9].

Despite recent advances in understanding the functions of MRAP2, little is known about the mechanism of the interaction of MRAP2 with the various GPCRs, except for data on melanocortin receptor 2 [10].

In this study, we characterize the formation of the MRAP2–PKR2 complex using biochemical techniques.

Specifically, we determine the PKR2 region involved in MRAP2 binding using yeast as a heterologous expression system. To identify the MRAP2 regions required for PKR2 binding, we express the C-terminal soluble domain of MRAP2 in *Escherichia coli* (*E. coli*). We also analyze prokineticin-stimulated G_αi_ signaling in CHO cell lines expressing PKR2. Finally, to determine the role of the prokineticin transduction system on MRAP2 expression regulation, we analyze MRAP2 levels in mice hypothalamic explants and adipocyte tissue.

## 2. Materials and Methods

### 2.1. Expression Constructs

For the constitutive expression of MYC-DDk-tagged human MRAP2 in mammalian cell lines, RC203668 plasmid was purchased from Origene (Fisher Scientific).

For the expression of MRAP2 in *Saccharomyces cerevisiae* (*S. cerevisiae*), the p413 MRAP2 plasmid was constructed. The cDNA of MRAP2 was amplified by PCR using a as template the RC203668 plasmid with the oligonucleotides MRAP2 *Bam*HI up and MRAP2 *Eco*RI dw and cloned into the p413 digested with *Bam*HI EcoRI under the control of the constitutive alcohol dehydrogenase promoter.

The procedure to construct pYESC3-PKR2, pYESC3-ΔN-PKR2 was previously described [11,12]. For *E. coli* expression of the carboxy-terminal domain of MRAP2, the pET 28 CT-MRAP2 construct was prepared. The DNA fragment of the C-terminal domain was amplified by PCR with gene-specific primers T70 *Bam*HI and MRAP2 *Eco*RI dw and cloned into pBluescript to obtain PBS-CT_MRAP2. The resulting plasmid was digested with *Bam*HI and *Eco*RI, and the fragment obtained was cloned into pET 28a. For the expression of the MRAP2 Δ131 deletion mutant, PBS CT-MRAP2 was digested with *Bam*HI and HindIII, and the resulting fragment was cloned into pET 28a to obtain pet 28-131CT-MRAP2; both vectors express the recombinant protein fused to a poly-his tag. All oligonucleotides used for the plasmid constructions are listed in Table 1.

### 2.2. Western Blot Assay

An equal amount of proteins were separated by electrophoresis, then transferred to a nitrocellulose membrane (TCM) and blocked in 5% non-fat milk BSA/Tris-buffered saline containing 0.5% Tween-20 (TBS-T pH 7.4) for 1 h at room temperature. Subsequently, the membranes were incubated overnight at 4 °C with the appropriate primary antibodies in the blocking solution. The primary antibodies used were rabbit anti-ERK_1/2_ (1:1000, Sant Cruz Biotechnology Inc, Santa Cruz, CA, USA, sc-153), mouse anti-pERK_1/2_ (1:1000, Cell Signaling Technology, Danvers, MA, USA, #9106S), mouse anti-STAT3 and rabbit anti-pSTAT3 (Tyr705) (1:1000, Invitrogen, Thermo Fisher Scientific, Milan, Italy, MA1-13,042 and 44-380G), monoclonal anti-polyhistidine-peroxidase (Sigma-Aldrich, Milan, Italy, A7058) and rabbit anti-MRAP2 polyclonal antibody (1:1000, Invitrogen-Thermo Fisher Scientific, Milan, Italy, PA5-113283). After extensive washing with T-TBS, membranes were incubated with the appropriate IgG HRP-linked secondary antibody for 1 h at room temperature. The immunoreactive signals were visualized using an enhanced chemiluminescence system (BM Chemiluminescence Blotting Substrate, Roche Diagnostics, Milan, Italy). After immunoblotting, digitized images of WB bands were acquired and the densitometric optical values were measured for each band using ImageJ software (version 1.47, http://imagej.nih.gov/ij/index.html, free software, accessed on 2 November 2021). The results were expressed as a percentage of control. The data were obtained from at least three independent experiments.

### 2.3. Ligand Production

In *P. pastoris*, PK2 was expressed in *P. pastoris* and purified, as described in Lattanzi et al., 2022 [13]. Briefly, the induction of the synthesis was carried out for 120 h in BMMY medium daily supplemented with 1% methanol. The crude cultures were centrifuged to remove the cells. Supernatants were diluted 1:5 and applied to a CM-Sephadex C-25 (Pharmacia) column in 20 mM BES, pH 7.0. The elution of the ligands was performed with 20 mM BES, pH 7.0/0.2 M NaCl. Recombinant ligands were pooled, dialyzed against 20 mM Tris–HCl (pH 7.0) buffer and analyzed in 18% (*w*/*v*) polyacrylamide SDS–PAGE gel.

### 2.4. Yeast Culture and Transformation

The *S. cerevisiae* strain used for the co-expression of PKR2 and MRAP2 or ΔN-PKR2 and MRAP2 was Cy12946 (MATa FUS1p-HIS3 GPA1Ga_i2(5)_ can1 far1D1442 his3 leu2sst2D2 ste14::trp1::LYS2 ste3D1156 tbt1-1 trp1 ura3). The transformation of Cy12946 yeast cells was performed according to the lithium acetate protocol [14]. Transformed yeast cells were grown at 30 °C to mid-log phase in an appropriate selection media containing 2% raffinose as the sole carbon source. Protein expression was induced by the addition of galactose to a final concentration of 20 g/L.

### 2.5. Preparation of Yeast Membrane Proteins

Yeast cells (1–2 × 10^7^ cells/mL) were collected by centrifugation, and membrane homogenates were prepared by a glass bead method, as previously described [11].

### 2.6. Yeast Membrane Proteins Cross-Linking by Bis-Sulfosuccinimidyl-Suberate (BS^3^)

A solution of 1 mM BS^3^ (Sigma) in 25 mM MOPS buffer pH 7.5 was freshly prepared. A total of 50 μL of membrane fraction (20 μg of protein) extracted from Cy12946 pYESC3 PKR2/p413 MRAP2 and Cy12946 pYESC3ΔN PKR2/ p413 MRAP2 was incubated together with BS^3^ for 30 min at room temperature. The reaction was stopped by the addition of 1 μL Tris–HCl 1.5 M. BS^3^ and control samples were analyzed by SDS–PAGE followed by immunoblotting with a monoclonal anti-His antibody (1:5000) and a polyclonal anti-MRAP2 antibody (1:1000) (Invitrogen-Thermo Fisher Scientific, Milan, Italy).

### 2.7. The Co-Precipitation of Yeast Membrane Extracts 

Solubilized membrane fractions co-expressing PKR2 and MRAP2 or ΔN-PKR2 and MRAP2 were co-precipitated by metal chelation chromatography using an Ni-NTA His-bind resin after batch purification under native conditions according to the manufacturer’s instructions (Novagen, Darmstadt, Germany). The eluates were fractionated by SDS–PAGE on a 12% gel and transferred to a polyvinylidene difluoride membrane. The membrane was coated with a monoclonal anti-His antibody (1:5000) or with a polyclonal anti-MRAP2 (1:1000) (Invitrogen-Thermo Fisher Scientific, Milan, Italy). 

### 2.8. Glutathione S-Transferase (GST) Pull-Down

The R2-GST proteins were obtained by a fusion of the 69 amino acids of the amino terminus of PKR2, with GST as described in [8,15]. After the IPTG-induced expression in *E. coli*, R2-GST fusion proteins were purified from cell extracts by affinity chromatography with glutathione-Sepharose beads (GE Healthcare, Hatfield, UK), according to the manufacturer’s instructions. Briefly, 50 μL slurry of glutathione-Sepharose beads equilibrated in buffer A (PBS, 1% Nonidet P 40, 1 mM EDTA supplemented with protease inhibitor) were incubated with 2 mL R2-GST lysate for 1 h at 4 °C with constant stirring. Beads with bound R2-GST were collected by centrifugation, washed extensively in buffer A, and incubated overnight at 4 °C with total membrane extracted from yeast expressing MRAP2 or with purified MRAP2 CT domain or purified Δ131-MRAP2 expressed in *E. coli.* Beads were washed, as described above, and bound proteins were eluted with GSH according to the procedure of GE Healthcare and analyzed by Western blotting with anti-His antibody (1:5000, Invitrogen-Thermo Fisher Scientific, Milan, Italy).

### 2.9. The Expression and Purification of the Recombinant Carboxy-Terminal Domain and Δ131-MRAP2 in E. coli

*E. coli* cultures expressing the carboxy-terminal domain and the Δ131-MRAP2 were grown at 37 °C until an optical density of 600 nm (OD_600_). Cultures were grown for an additional 4 h at 28 °C in the presence of isopropyl-β-D-1-thiogalactopyranoside (IPTG) 0.1 mM. The C-terminal domain of MRAP2 and the Δ131 -MRAP2 were purified using a His-tagged protein purification kit (Novagen, Darmstadt, Germany). The purified recombinant proteins were resolved by SDS-PAGE and analyzed by Western blotting with anti-His antibody (1:5000, Invitrogen-Thermo Fisher Scientific, Milan, Italy).

### 2.10. Blue Native PAGE

A 10× loading dye (5% *w*/*v* Coomassie Blue G-250, 500 mM 6-aminohexanoic acid) was added to 20 μg of purified C-terminal MRAP2 domain and loaded on Native PAGE Novex 3% to 16% Bis–Tris gels (Thermo Fisher, Life Technologies, Inc, Milan, Italy). After electrophoresis at 0 °C (using buffers and conditions specified by the manufacturer and reported in [16]), proteins were transferred to a polyvinylidene fluoride membrane (60 min at 15 V), which was pre-wetted in methanol for 30 s and then soaked in transfer buffer for several minutes [16]. The membrane was decolorized in methanol for 3 min, rinsed in T-TBS pH 7.4, blocked, and immunoblotted with monoclonal anti-His (1:5000, Invitrogen-Thermo Fisher Scientific, Milan, Italy).

### 2.11. CHO-PKR2 Cell Culture, Transfection, and Stimulation

CHO cells (Chinese hamster ovary) stably expressing human PKR2 were plated at a density of 4 × 10^5^ per well in 6-well plates and allowed to grow until they reach 70–80% of confluency. Cells were grown in Dulbecco’s Modified Eagle Medium/Nutrient mixture F-12 Ham (DMEM/F12, Sigma Aldrich) containing 100 U/mL penicillin/streptomycin, 10% Fetal Bovine Serum (FBS), 2 mM L-glutamine, and G418 200 ng/mL (Sigma-Aldrich, Milan, Italy) at 37 °C and 5% CO_2_. Transient transfection with plasmid DNA of MRAP2 was performed in the presence of Lipofectamine 2000 (Invitrogen Life Technologies, Milan, Italy) according to the manufacturer’s instructions. Briefly, cells were incubated with lipo-DNA complex for 24 h at 37 °C and 5% CO_2_. The lipo–DNA complex was obtained by incubating 2 μg of plasmid DNA for MRAP2 with Lipofectamine 2000 diluted in Opti-MEM (Invitrogen, Thermo Fisher Scientific) for 20 min at room temperature. A total of 24 h after transfection cells were serum-starved and stimulated with PK2 (100 nM) for 10 min and 1 h at 37 °C, 5% CO_2_. At the end of the incubation period, cells were lysed in RIPA lysis buffer containing a protease inhibitor cocktail (1% *v*/*v*) (Sigma Aldrich), quantified, and used for WB analysis. 

### 2.12. Adipose Tissue and Hypothalamus Explants

Epididymal adipose tissue was collected from 4- and 10-month-old WT (n = 4) and PKR1-KO (n = 4) mice, rapidly frozen on dry ice and stored at −80 °C until RNA or protein extraction. 

Hypothalami were extracted from 4-month-old WT male mice (n = 12) immediately following decapitation, placed in oxygenated ice-cold Krebs–Ringer bicarbonate buffer (pH 7.4; 4 °C) and successively cut into 300 μm thick slices using a vibratome. Hypothalamic slices were incubated in Dulbecco’s Modified Eagle Medium (DMEM, Merck Sigma-Aldrich, Milan, Italy) supplemented with 1% of Fetal Bovine Serum (FBS, Merck Sigma-Aldrich, Milan, Italy) at 37 °C, 5% CO_2_ and 95% O_2_ for 2 h, then serum-starved and treated with PK2 (100 nM) or PK2 plus WP1066 a STAT3 inhibitor (Merck KGaA) for 1 h. WP1066 (5 μM) was added 1 h before PK2 addition. At the end of the incubation period, the explants were quickly frozen on dry ice and stored at −80 °C until RNA or protein were extracted. Total cell protein was extracted using RIPA lysis buffer containing a protease inhibitor cocktail (1% *v*/*v*) (Sigma-Aldrich, Milan, Italy), quantified, and used for WB analysis.

Total RNA was isolated from adipose tissue and hypothalamic explants using Trizol reagent (Invitrogen, Thermo Fisher Scientific) according to the manufacturer’s instructions. After determining its yield and purity by spectrophotometric absorbance at 260 and 280 nm, RNA was used for real-time PCR analysis.

### 2.13. Real-Time PCR

One microgram of extracted RNA was reverse transcribed using SensiFAST cDNA Synthesis Kit (Meridian Bioscience, Milan, Italy) and the cDNA obtained was used to perform real-time PCR. All reactions were performed in triplicate using 25 ng of cDNA, iQ SYBR Green Supermix (Meridian Bioscience, Milan, Italy), and specific mouse primers (Table 1). The reaction conditions were as follows: 10 min at 95 °C for polymerase activation, followed by 40 cycles at 95 °C (denaturation), 30 s at 60 °C (annealing) and 30 s at 72 °C (extension). A reaction mixture without cDNA was used as a control. The Ct value of MRAP2 was normalized to the Ct value of the endogenous control, glyceraldehydes-3-phosphate dehydrogenase (GAPDH), using the comparative Ct method (2^−ΔΔCt^).

### 2.14. Data Analysis

Data were analyzed using GraphPad Prism 6 for Windows by Student *t*-test or One-Way ANOVA followed by Tukey’s post-test when appropriate. Differences were considered significant at *p* < 0.05.

## 3. Results 

### 3.1. MRAP2 Inhibits PKRs G_i_ Coupling

MRAP2 is an important regulator of energy homeostasis that increases MC4R activation and inhibits PKR signaling [4]. In CHO cells stably transfected with PKR1 or PKR2, PK2 is able to activate ERK_1/2_ and STAT3 pathways [7,13,15,17,18], and this activation is abrogated by pretreatment with the G_αi_ inhibitor pertussis toxin. We, therefore, examined ERK_1/2_ and STAT3 phosphorylation in CHO-PKR2 cells in the presence or absence of MRPA2, after PK2 incubation. 

The results show that when PK2 binds to PKR2, it induces a strong ERK_1/2_ and STAT3 phosphorylation. In contrast, in the presence of MRAP2, the effect of PK2 is significantly reduced. This suggests that the presence of MRAP2 reduces PK2-induced PKR2 activation and impairs G_i_ coupling (Figure 1). 

### 3.2. Identification of PKR2 Region Involved in MRAP2 Binding in S. cerevisiae

Recent data show that MRAP2 is able to inhibit PKR2 glycosylation of Asparagine at position 27 of the N-terminal extracellular region [18]. The absence of glycosylation results in the impaired localization of PKR2 to the plasmatic membrane. To determine whether MRAP2 prevents glycosylation by directly binding to the N-terminal region of PKR2, we expressed a deletion mutant of the N-terminal region (ΔN-PKR2) in *S. cerevisiae*. PKR2 and ΔN-PKR2 terminal-tagged His (PKR2-His; ΔN-PKR2His) were expressed with MRAP2 in yeast Cy12946, and total membrane proteins were purified by Ni-NTA chromatography and subjected to Western blotting analysis. As shown in Figure 2, MRAP2 binds only to fulfill length PKR2 as indicated by the signal obtained with both the anti-His and -MRAP2 antibodies. No signal was detected with anti-MRAP2 (Figure 2, lane 2) on the ΔN-PKR2 /MRAP2 membrane extract. This result was also confirmed by GST pull-down experiments. The fragment encoding the N-terminal region of PKR2 was inserted into the frame of the GST cDNA (PKR2-NT). Lysate from cells expressing the GST fusion protein PKR2-NT was incubated with glutathione-Sepharose resin in the presence of membrane proteins purified from *S. cerevisiae* cells expressing MRAP2. As shown in Figure 2B, PKR2-NT is able to pull-down MRAP2, demonstrating their interaction. Conversely, GST alone is not able to pull-down MRAP2. Further evidence for the MRPA2-NT-PKR2 interaction was provided by crosslinking experiments. Total proteins from *S. cerevisiae* cells co-expressing PKR2/MRAP2 or ΔN-PKR2/MRAP2 were fractionated into the membrane and cytosolic proteins and exposed to BS^3^ cross-linker. Proteins were separated using SDS−PAGE, blotted, and then probed with the anti-MRAP2 antibody (Figure 3; Appendix A). A distinct band was detected in the lane corresponding to the membrane protein extracted from PKR2/MRAP2 expressing yeast cells. The size of the band indicates that it is the PKR2/MRAP2 complex obtained by cross-linking. Similar results were observed in three independent replicates of this experiment. These results, obtained by cross-linking, confirm that the PKR2 N-terminal region is essential for MRAP2 binding.

### 3.3. Biochemical Analysis of the Interaction between the C-Terminal Region of MRAP2 and Prokineticin Receptor 2

The human MRAP2 gene consists of four exons and three introns. The first exon contains a 5′ untranslated sequence (UTR); the second exon contains a portion of the 5′ UTR sequence and a region encoding the first 44 amino acids; the third exon encodes a region comprising from the amino acid at position 45 to the amino acid at position 77; and the fourth exon encodes for the C-terminal cytosolic domain consisting of 129 amino acids and the 3′ UTR sequence (Figure 4A).

To identify the MRAP2 region, which is important for PKR2 interaction, we generated an MRAP2 deletion mutant containing only the C-terminal region encoded by the fourth exon (CT-MRAP2) and a deletion mutant containing only the region extending from the residue at position 78 to the residue at position 131 (131CT-MRAP2) (Figure 4B).

The MRAP2 mutants were produced in *E. coli.* The molecular mass of the purified proteins fused to His TAG was determined by SDS-PAGE and was approximately 20 kDa and 10 kDa, respectively, which is close to the values derived from the gene sequence. The identity of the purified proteins was also confirmed by Western blotting with an anti-His-tag antibody (Figure 4C).

Electrophoresis analysis showed that the CT-MRAP2 protein maintains the ability of dimerization, and that the protein also migrates as a dimer under denaturing conditions as previously described [1].

To define the ability of the CT-MRAP2 region to form higher-order oligomers better, we used blue native PAGE. Based on the molecular weights of the bands, we were able to detect the presence of structures that are most likely dimers. In the blue native PAGE, the detergent affects the mobility of the protein complex through the gel matrix, so the prediction of the molecular weights is only approximatively (Figure 4D).

The interaction of the amino-terminal region of PKR2 with CT-MRAP2 and 131CT-MRAP2 was investigated by glutathione S-transferase (GST) pull-down experiments. Lysate from cells expressing the GST fusion protein PKR2-NT was incubated with glutathione-Sepharose resin in the presence of the purified CT-MRAP2 and 131CT-MRAP2 domains. As shown in Figure 4E, PKR2-NT is able to pull-down MRAP2 domains, demonstrating the ability of PKR2 to interact with both MRAP2 deletion mutants. Conversely, GST alone is not able to pull-down MRAP2. Moreover, it appears from the gel that PKR2 is more likely to bind the dimeric form of CT. 

### 3.4. Modulation of MRAP2 Expression by the Prokineticin System

Because MRAP2 is involved in energetic homeostasis, we examined the modulation of MRAP2 expression by the prokineticin system both in peripheral tissues, adipose tissue, and in the central nervous system, hypothalamus.

In adipose tissue from wild-type (WT) and PKR1 knock-out (KO) mice, the expression of MRAP2 is up-regulated in adipocytes from PKR1 KO mice at 10 months of age at both mRNA and protein levels (Figure 5A,B).

In ex vivo hypothalamic explants, PK2 treatment significantly reduces MRAP2 expression and pre-incubation with a STAT3 inhibitor restores MRAP2 expression to physiological levels (Figure 5C,D).

## 4. Discussion

MRAP2 is a single transmembrane protein expressed mainly in the stomach and endocrine glands, in the hypothalamus, and in adipocytes [3]. It is found on the cell surface and the reticulum membrane [1]. MRAP2 is essential for energy homeostasis, as it modulates the activity of hypothalamic neurons interacting with various systems, such as the melanocortin-4 receptor and orexin, ghrelin, and prokineticins [3,19]. Preliminary studies have shown that MRAP2 is able to inhibit prokineticin receptor signaling by controlling the levels of cAMP and IP3 [4]. We, therefore, investigated whether MRAP2 is able to regulate PKR2 signaling through G_αi_ coupling. The reduction in STAT3 and ERK_1/2_ activation in CHO cells expressing PKR2 and exposure to PK2 showed that MRAP2 also negatively modulates G_αi_ coupling. 

MRAP2 impairs PKR2 localization to the plasma membrane and inhibits PKR2 glycosylation in the correspondence of Asparagine at position 27 of the N-terminal extracellular region [20]. To determine whether MRAP2 prevents glycosylation by directly binding the N-terminal region of PKR2, we expressed a deletion mutant of N-terminal region (ΔN-PKR2) and PKR2 in *S. cerevisiae*. Using biochemical techniques, such as co-precipitation, GST pull-down and cross-linking, we demonstrated that MRAP2 binds specifically to PKR2, but does not interact with the ΔN mutant. These experiments showed that the N-terminal region of PKR2 is the essential part for interaction with MRAP2. This result may explain the differential ability of MRAP2 to regulate PKR1 and PKR2 in signal transduction and trafficking [4], because the two receptors share a high sequence identity excluding only the N-terminal region. Subsequently, it was identified the MRAP2 region which is critical for PKR2 binding. The C-terminal region of MRAP2 is conserved in evolution and required for the regulation of orexin, ghrelin, and PKRs receptors [5,21]. We expressed two different portions of the C-terminal domain of MRAP2 in *E. coli*: the entire C-terminal domain (76–205) and a limited region between amino acids at position 76 to the amino acid at position 131 (131CT domain). We demonstrated that the C-terminal domain of MRPA2 does not lose the ability to form a dimer structure. This is consistent with a dual topology of the protein and evidence that the C-terminal region is involved in parallel dimer formation [22]. Experiments with GST pull-down show that the CT-domain contains the minimal regions necessary for PKR2 binding. Finally, we have demonstrated that the prokineticin system regulates the expression of MRAP2. We examined MRAP2 expression by real-time PCR and Western blotting in adipocytes from wild-type and PKR1 knock-out mice. In mice lacking the gene encoding PKR1, the expression of MRAP2 is strongly increased at both transcriptional and translational levels. The increase in MRAP2 levels was found only in 10-month-old PKR1-KO mice. The PKR1-KO mice adipocytes are considered a model for obese and diabetic phenotype only when mice have reached 10 months of age because they exhibit a hypoxic state, a reduced glucose tolerance, low expression of insulin receptors, and low levels of insulin-stimulated Akt phosphorylation [9,23].

This effect could be mediated by the upregulation of PPARɣ in adipocytes of PKR1-KO mice [9], as PPARɣ is a transcriptional regulator of MRPA2 expression [24]. 

We incubated mice hypothalamic explants with PK2 or PK2 in the presence of STAT3 inhibitor and we examined the expression of MRAP2 at the transcriptional and translational levels. Treatment with PK2 results in a significant reduction in the expression of MRAP2, and this reduction in expression is absent when explants were previously treated with the STAT3 inhibitor. This result suggests that STAT3 is involved in the regulation of MRAP2 expression.

## 5. Conclusions

PKR2 has two putative glycosylation sites in the N-terminal region at position 7 and position 27. It was demonstrated that, by site-direct mutagenesis, glycosylation at position 27 is essential for the trafficking of PKR2 at the plasma membrane, and that this post-translational modification is absent in the presence of MRAP2 [20]. Indeed, MRAP2 prevents glycosylation and blocks PKR2 in the endoplasmic reticulum [20]. Our results suggest that MRAP2 binds to the N-terminal region of PKR2, preventing glycosylation and consequently the correct localization of the receptor. We also identified a specific region of MRAP2 that is critical for the dimer formation and interaction with PKR2.

There is sufficient evidence that the C-terminus of MRAP2 is the most important domain for regulating GPCRs, but there is very little information about the structure of MRAP2, even if it seems that it does not contain an ordered secondary structure [3]. The hypothesis is that the lack of a fixed structure allows MRAP2 to interact with multiple protein partners. It is possible that MRAP2 adopts a more rigid conformation in the presence of different GPCR receptors. This confirms that the MRAP2 region involved in the interaction with receptors is as GPCR-specific as the regions involved in the regulation of traffic and activity [24].

## Figures and Tables

**Figure 1 biomolecules-12-00474-f001:**
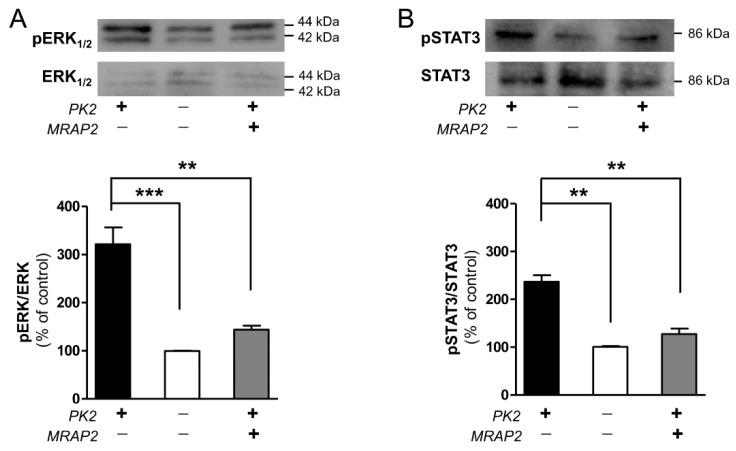
Analysis of ERK_1/2_ and STAT3 phosphorylation in CHO-PKR2 cells in the presence of MRAP2. Representative Western blots and densitometric plots showing pERK_1/2_ and ERK_1/2_ (**A**) and p-STAT3 and STAT3 (**B**) protein levels after 10 min and 1 h treatments, respectively, with PK2 100 nM. The bar graph data are presented as a ratio of pERK_1/2_ /ERK_1/2_ and pSTAT3/STAT3 and plotted as % of increase respect to non-stimulated cells. Bar plots indicate the means ± SEM obtained from the three experimental conditions. One-way ANOVA was used for statistical evaluation, followed by Tukey’s post-test for multiple comparisons ** *p* < 0.01, *** *p* < 0.001.

**Figure 2 biomolecules-12-00474-f002:**
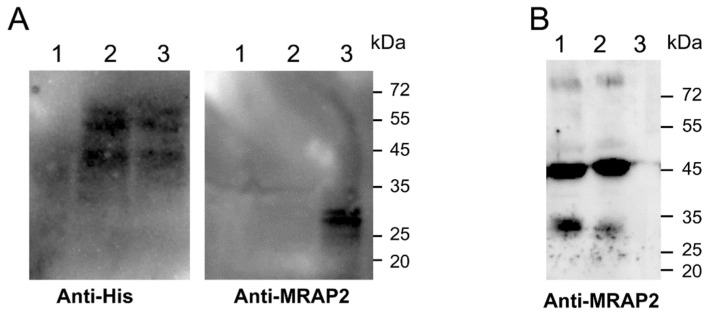
Interaction of MRAP2 with the N-terminal region of PKR2. (**A**) Membranes from yeast co-expressing MRAP2 with PKR2 or ΔN-PKR2 were co-precipitated using Ni-NTA His-bind resin and were resolved by 12% SDS-PAGE. The immunoblots were probed with anti-His (left panel) and anti-MRAP2 (right panel) antibodies. Lane 1: Cy12946, lane 2: ΔN-PKR2, and lane 3: PKR2. (**B**) The GST fusion protein PKR2-NT was used to pull-down MRAP2, and the elution solutions obtained were resolved 15% SDS-PAGE analyzed by Western blotting using an anti-MRAP2 antibody. Lane 1: input; lane 2: PKR2-NT eluate; and lane 3: GST eluate.

**Figure 3 biomolecules-12-00474-f003:**
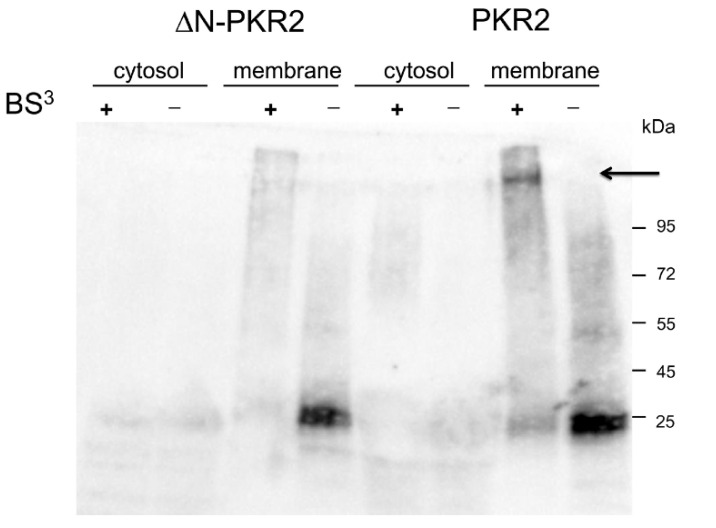
Cross-linking of MRAP2 with PKR2 or ΔN-PKR2 receptors. Membrane and cytosolic proteins prepared from *S. cerevisiae* cells expressing the PKR2-MRAP2 and ΔN-PKR2 -MRAP2 were incubated in the presence (+) or absence (−) of BS^3^. Proteins were immunoblotted and probed with an anti-MRAP2 antibody. The arrow indicates the complex PKR2/MRAP2.

**Figure 4 biomolecules-12-00474-f004:**
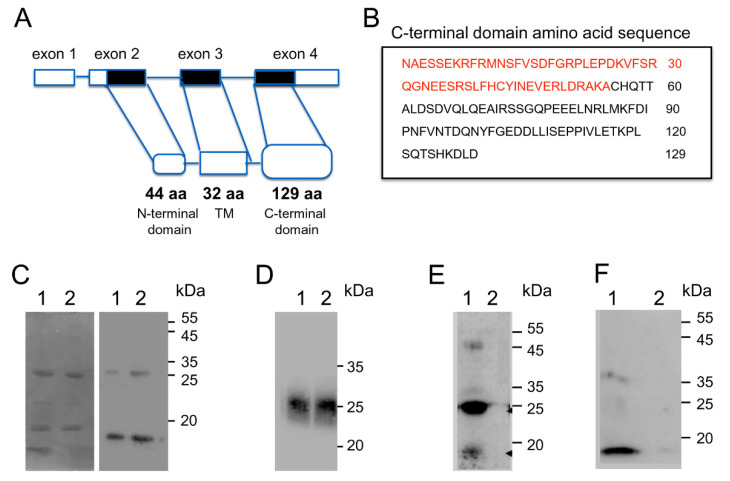
Analysis of MRAP2 regions that bind PKR2. (**A**) Scheme of human MRAP2 gene. Exon coding sequences are indicated as black bars and the untranslated sequences are shown as white bars. (**B**) In the box, the amino acid sequence of the C-terminal MRAP2 domain; the sequence present in the 131CT-MRAP2 domain is indicated in red. (**C**) SDS PAGE and Western blot analysis of CT-MRAP2 domain expression in *E. coli*; lane 1: 5 μg; lane 2: 10 μg. (**D**) Western blot analysis of native PAGE of CT-MRAP2 domain expression in *E. coli*; lane 1: 10 μg; lane 2: 15 μg. (**E**) The GST fusion protein and PKR2-NT were used to pull-down the CT-MRAP2 domain. Solutions obtained by elution were analyzed by Western blotting: lane 1; PKR2-NT eluate; lane 2: GST eluate. (**F**) The GST fusion protein and PKR2-NT were used to pull-down the 131CT-MRAP2 domain. Solutions obtained by elution were analyzed by Western blotting: lane 1; PKR2-NT eluate; lane 2: GST eluate.

**Figure 5 biomolecules-12-00474-f005:**
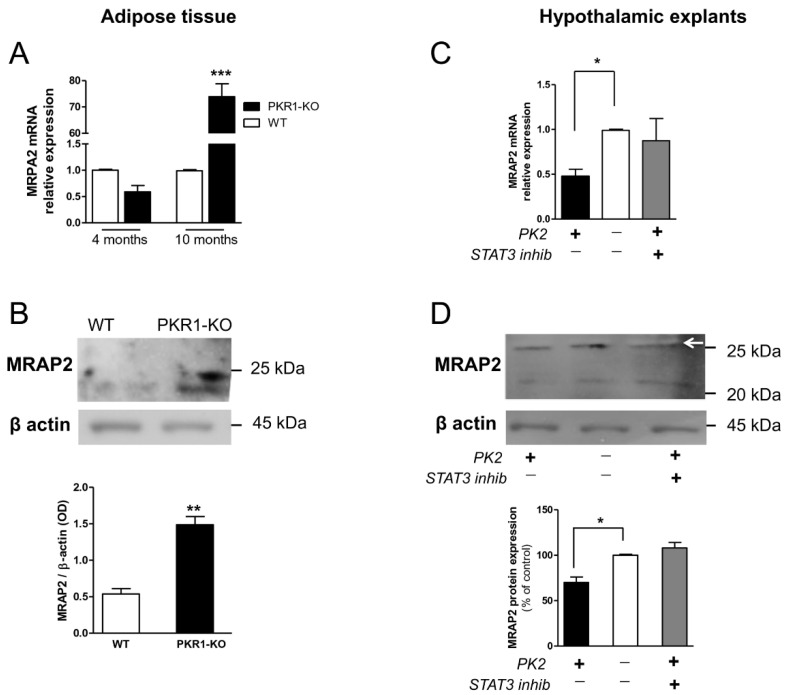
MRAP2 expression in adipocytes and hypothalamic explants. MRAP2 levels in adipose tissue were analyzed by (**A**) real-time RT-PCR and (**B**) Western blot analysis. MRAP2 levels in hypothalamus explants analyzed by (**C**) real-time RT-PCR and (**D**) Western blot analysis (arrow). Data are expressed as the mean ± SEM of three separate experiments (n = 4 per group). Statistical analyses were performed using Student’s *t*-test where ** *p* < 0.01; *** *p* < 0.001 vs. WT (**A**,**B**) and one-way ANOVA followed by Tukey’s post-test, * *p* < 0.05 (**C**,**D**).

**Table 1 biomolecules-12-00474-t001:** Oligonucleotides used in the study.

Oligonucleotide	Sequence
MRAP2 *BamH*I up	5′-AAG GAT CCA TGTCCGCCCAGAGG-3′
MRAP2 *EcoR*I dw,	5′-AAGAATTCTTAAACCTTATCGTC-3′
T70 BamHI	5′-GGATCCACCAAGACAGGAGCCCCA-3′
GAPDH fw	5′-GCC AAG GCT GTG GGC AAG GT-3′
GAPDH rv	5′-TCT CCA GGC GGC ACG TCA GA-3′

## Data Availability

The data presented in this study are available on request from the corresponding author.

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
