# Peer review of "Identification of Regions Involved in the Physical Interaction between Melanocortin Receptor Accessory Protein 2 and Prokineticin Receptor 2"

_biomolecules, 2022, doi:10.3390/biom12030474_

Round 1
Reviewer 1 Report
In the present MSS entitled “Identification of regions involved in physical interaction be-2 tween Melanocortin Receptor Accessory Protein 2 and Proki-3 neticin Receptor 2” Fullone et al., described the changes in trafficking and signal transduction of several G-protein-coupled receptors (GPCRs) in presence of Melanocortin Receptor Accessory Protein 2 (MRAP2). The approach used in the present study is good and concept is very clear. MSS in general is well written, Methodology is well described whereas introduction is poorly written. Several points which are missing in introduction are scattered in other part of MSS such as in results section. Authors obtained very good results but the quality of illustration is poor (discussed below). Also, results are not discussed to the points. These few points distract smooth reading of the MSS.
Some of the comments are as following
In first sentence of abstract authors stated “several G-protein-coupled receptors (GPCRs)” but there are not several GPCRs studied here, it should be corrected.
In discussion section line 343 authors stated “a second goal was” but was the first goal is missing.
In cross linking experiment authors obtained convincing results but did not discuss in text.
In the discussion section line 362 to 367 information provided does not fit there.
Methods section, Western blot assay is not well written, please provide all the details.
Authors determined ERK1/2 and that should be corrected in text. Figure is not showing strong phosphorylation as stated in result section. If authors dissected out ERK isoform specific changes then results will make some significant observation. In contrast, pSTAT3 status is quite convincing.
The absence of glycosylation results in impaired localization of PKR2 to the plasmatic membrane. This statement is not clear. What is this plasmatic?
Result section 3.1, first three line with references can be moved to introduction.
The use of term GPCRs in respect to the binding to MRAP2 interaction is very broad and does not fit here.
Through out the MSS not a single Western blot is supporting results properly. It is very difficult to judge how densitometry was done when there is no proper band is seen. Importantly, no molecular weight marker is run along with sample.
In figure 5 A, the loss of MRPA2 mRNA in adipose tissue at 4 month and significant increased at 10 months is not discussed and Western blot in Panel B is of poor quality. Same is true for Panel D.
MSS need careful reading
Author Response
Some of the comments are as follows:
- In first sentence of abstract authors stated “several G-protein-coupled receptors (GPCRs)” but there are not several GPCRs studied here, it should be corrected.
The opening sentence has been inserted as a useful sentence to outline the general role of MRAP2.
- In discussion section line 343 authors stated “a second goal was” but was the first goal is missing.
Thanks for the suggestion we changed the sentence.
- In cross linking experiment authors obtained convincing results but did not discuss in text.
Thanks for the suggestion. We added a comment in the results.
4.In the discussion section line 362 to 367 information provided does not fit there.
We have moved the phrase into the introduction.
- Methods section, Western blot assay is not well written, please provide all the details.
Thanks to the suggestion we have changed the paragraph on the materials and methods.
- Authors determined ERK1/2 and that should be corrected in text. Figure is not showing strong phosphorylation as stated in result section. If authors dissected out ERK isoform specific changes then results will make some significant observation. In contrast, pSTAT3 status is quite convincing.
Thanks for the suggestion, we introduced ERK 1/2 in the figure and checked the graphics.
The graph shows the data obtained from the quantification of the bands and the result is comparable to that already obtained for ERK activation by PK2 in other systems (Lattanzi et al., 2018).
- The absence of glycosylation results in impaired localization of PKR2 to the plasmatic membrane. This statement is not clear. What is this plasmatic?
Sorry for the mistake. We intended plasma membrane.
- Result section 3.1, first three line with references can be moved to introduction.the use of term GPCRs in respect to the binding to MRAP2 interaction is very broad and does not fit here.
Thanks for the suggestion. We have thought about it, and we decided to leave the sentence in the results because it allows us to explain why we focused on ERK and STAT activation to analyze the PKR2 pathway through Gai.
- Throughout the MSS not a single Western blot is supporting results properly. It is very difficult to judge how densitometry was done when there is no proper band is seen. Importantly, no molecular weight marker is run along with sample.
Westerns made using proteins extracted from yeast strains do not allow to obtain well-defined bands. However, the analysis of the intensity of the bands was done in a rigorous manner using ImageJ software (version 1.47, http://imagej.nih.gov/ij/index.html, free software).
- In figure 5 A, the loss of MRPA2 mRNA in adipose tissue at 4 month and significant increased at 10 months is not discussed and Western blot in Panel B is of poor quality. Same is true for Panel D.
Thanks. We added a phrase in the discussion to comment that the obese phenotype in PKR1-KO mice is reached only in mice 10 months old.
- MSS need careful reading.
The work was all checked in order to improve the form.

Reviewer 2 Report
This work studies the interaction of MRAP2 and PKR2 and aims to demonstrate that the C-terminal region of MRAP2 binds the N-terminal region of PKR2. This is an interesting study which would further characterize the previous finding by another group that MRAP2 prevents PKR2 glycosylation resulting in receptor retention in the ER.
Here are several concerns with this manuscript:
- The paper contains numerous typos and grammatical errors. A thorough proofreading is required.
- This paper heavily relies on an anti-MRAP2 antibody that has not been properly validated. Validation using cells transfected with and without MRAP2 for in vitro use and using tissues from MRAP2 KO mice for validation for tissue lysates is a must to provide confidence in this reagent.
- PKR2 signaling through Gas and Gaq/11 is well documented. Signaling through Gai less so. For this reason, the experiments in figure 1 need to be properly controlled and should include a pertussis toxin control to verify the involvement of Gi/o in the signaling pathway studied. The source of multiple reagents is missing, for example, where did the author get PK2 from?
- Figure 2A. The blot shown is extremely dark making it very difficult to see bands and determine if non-specific bands are present. Also, PKR2 and DN-PKR2 appear to run at the same size which needs to be explained because it does not really make sense. DN-PKR2 should be running faster than the full size.
- Figure 3: This crosslinking experiment is confusing. First, the results in DN-PKR2 and PKR2 are essentially identical. Also, since this is not a PKR2 pulldown, the only thing this experiment demonstrates is that MRAP2 can be crosslinking with itself or other proteins to form higher complexes. It does not prove in any way that it is crosslinked to PKR2. The faint band pointed out by the authors may be complexes between MRAP2 and PKR2 but this experiment does not prove it. Would the result be the same in cells expressing only MRAP2?
- Figure 4: As mentioned by the authors in the discussion, the C-terminal region of MRAP2 is not organized and for that reason, when produced in E.coli and purified, it has a high tendency to aggregate. Consequently, the idea that C-terminal regions can form dimers based on the size it runs at on a gel is not an accurate conclusion. This is likely just the consequence of aggregation of the protein when it is not in its proper environment. Authors should perform more assays like light scattering on samples after size exclusion chromatography to determine the level of aggregation of MRAP2.
- Figure 5: It is not clear why the authors decided to look at PKR1 KO mice when the paper focusses on PKR2. Where are the PKR1-KO mice from? Here again the western blots are messy and make interpretation difficult. MRAP2 antibody should be tested on samples from MRAP2-KO mice. No band corresponding to MRAP2 is present in the blot of fig 5D. the size of the error bar in the STAT3 inhibitor qPCR samples in figure 5C precludes making conclusions on the role of STAT3-mediated regulation of MRAP2 expression. These experiments will require increased sample numbers for qPCR and better western blot data.
Author Response
Here are several concerns with this manuscript:
- The paper contains numerous typos and grammatical errors. A thorough proofreading is required.
The work was all checked in order to improve the form.
- This paper heavily relies on an anti-MRAP2 antibody that has not been properly validated. Validation using cells transfected with and without MRAP2 for in vitro use and using tissues from MRAP2 KO mice for validation for tissue lysates is a must to provide confidence in this reagent.
The antibody is a commercial antibody specific for humans and mice. Nevertheless, the antibody was also tested with total protein extracts obtained from mouse skin and colon, where the MRAP2 protein is not expressed, and we did not obtain a non-specific binding to other proteins, validating the specificity of the antibody.
- PKR2 signaling through Gas and Gaq/11 is well documented. Signaling through Gai less so. For this reason, the experiments in figure 1 need to be properly controlled and should include a pertussis toxin control to verify the involvement of Gi/o in the signaling pathway studied.
The Gai signaling is well documented both in vivo and in cell cultures.
The PKR2 and S1PR1receptors are the only two cases of GPCR receptors able to activate of Jak2/Stat3 signaling by Gα i/o coupling (Xin et al.; 2013). Moreover, we have already published the STAT3 activation in CHO expressing PKRs cells. In the same way, is well documented that the activation of PKRs induces ERK phosphorylation by Gα/o coupling. PK2-induced ERK phosphorylation in human monocytes is inhibited by pertussis toxin (Le Couter et al., 2004) suggesting involvement of the Gi proteins. Moreover, we have already published the ERK activation in CHO-expressing PKRs cells.
We already showed this evidence in 3.1 paragraph.
- Xin, H., Lu, R., Lee, H., Zhan., et al., 2013. G-protein-coupled receptor agonist Bv8/prokineticin-2 and STAT3 protein form a feed-forward loop in both normal and malignant myeloid cells. J. Biol. Chem. 288, 13842–13849. http://dx.doi.org/10.1074/jbc.M113.450049.
- LeCouter J, Zlot C, Tejada M, Peale F, Ferrara N. Bv8 and endocrine gland-derived vascular endothelial growth factor stimulate hematopoiesis and hematopoietic cell mobilization. Proc Natl Acad Sci USA 101: 16813–16818, 2004. doi:10.1073/pnas.04076971014.
- The source of multiple reagents is missing, for example, where did the author get PK2 from?
Thanks for the suggestion. PK2 was produced using Pichia pastoris as a heterologous expression system with a procedure developed by us several years ago. We have introduced this information into the text in the section “material and methods”.
- Figure 2A. The blot shown is extremely dark making it very difficult to see bands and determine if non-specific bands are present. Also, PKR2 and DN-PKR2 appear to run at the same size which needs to be explained because it does not really make sense. DN-PKR2 should be running faster than the full size.
The DN mutant presents a deletion of about 40 amino acids, so it has a putative molecular weight of about 31 kDa while the wt receptor has a monomer weight of about 35 kDa This difference in molecular weight is not appreciable in the separation of PKR2, both in monomeric and dimeric form, by electrophoresis using 12% SDS PAGE.
- Figure 3: This crosslinking experiment is confusing. First, the results in DN-PKR2 and PKR2 are essentially identical. Also, since this is not a PKR2 pulldown, the only thing this experiment demonstrates is that MRAP2 can be crosslinking with itself or other proteins to form higher complexes. It does not prove in any way that it is crosslinked to PKR2. The faint band pointed out by the authors may be complexes between MRAP2 and PKR2 but this experiment does not prove it. Would the result be the same in cells expressing only MRAP2?
In the lane corresponding to proteins treated with the cross-linker obtained by cells co-expressing MRAP2 and the mutant DN-PKR2, a low band of about 25 kDa is observed. The experiment was repeated also with proteins treated with the cross-linker obtained by cells co-expressing MRAP2 alone and the result was the same (data not shown). This demonstrates that MRAP2 alone is not able to give high molecular weight complexes in the presence of the cross-linker. We have introduced an arrow in the figure to indicate the band corresponding to the complex present only in the presence of PKR2 wt and MRAP2 to facilitate the understanding of the experiment.
- Figure 4: As mentioned by the authors in the discussion, the C-terminal region of MRAP2 is not organized and for that reason, when produced in E.coli and purified, it has a high tendency to aggregate. Consequently, the idea that C-terminal regions can form dimers based on the size it runs at on a gel is not an accurate conclusion. This is likely just the consequence of aggregation of the protein when it is not in its proper environment. Authors should perform more assays like light scattering on samples after size exclusion chromatography to determine the level of aggregation of MRAP2.
The protein expressed in E. coli corresponding to the C-terminal domain of MRAP2 was purified by affinity chromatography and then analyzed using electrophoresis in the presence of SDS. Firstly, we expressed the protein, as a cytosolic protein, in E. coli deleting the transmembrane element. The protein, produced in large amounts and in a soluble form in E.coli, was analyzed by SDS-PAGE electrophoresis. The result reveals the presence of a band corresponding to the dimer. This result confirms the capacity of the protein to form a stable SDS-resistant dimer. Moreover, previous studies with the aim to define the regions involved in the formation of the dimer, demonstrated that deletion mutant proteins, continue to have the ability to form stable SDS-resistant dimers.
The dimeric structure of the CT-MRAP protein could be broken by treating the protein before electrophoresis with UREA. In this case, a single band, corresponding to the monomer able to bind the antibody against the HIS TAG present at the protein N-terminal region, is evident.
- Figure 5: It is not clear why the authors decided to look at PKR1 KO mice when the paper focuses on PKR2. Where are the PKR1-KO mice from? Here again, the western blots are messy and make interpretation difficult. MRAP2 antibody should be tested on samples from MRAP2-KO mice.
We used PKR1-KO 40 week old mice because it is considered a model for an obese and diabetic phenotype. Here are the references:
- C. Szatkowski, J. Vallet, M. Dormishian, N. Messaddeq, P. Valet, M. Boulberdaa, D. Metzger, P. Chambon, C.G. Nebigil, Prokineticin receptor 1 as a novel suppressor of preadipocyte proliferation and differentiation to control obesity, PLoS ONE 8 (12) (2013) 81175, https://doi.org/10.1371/journal.pone.0081175.
- M. Dormishian, G. Turkeri, K. Urayama, T.L. Nguyen, M. Boulberdaa, N. Messaddeq, G. Renault, D. Henrion, C.G. Nebigil, Prokineticin Receptor-1 Is a New Regulator of Endothelial Insulin Uptake and Capillary Formation to Control Insulin Sensitivity and Cardiovascular and Kidney Functions, J. Am. Heart. Assoc. 2 (5) (2013), e000411, https://doi.org/10.1161/JAHA.113.000411.
- No band corresponding to MRAP2 is present in the blot of fig 5D. the size of the error bar in the STAT3 inhibitor qPCR samples in figure 5C precludes making conclusions on the role of STAT3-mediated regulation of MRAP2 expression. These experiments will require increased sample numbers for qPCR and better western blot data.
Now we used an arrow to indicate the band corresponding to MRAP2. We increased the sample numbers by repeating the experiment. Now, the size of the error bar in the STAT3 inhibitor qPCR samples is reduced and so, the result regarding the role of STAT3-mediated regulation of MRAP2 expression is clearer. Thanks for the suggestion.
Round 2
Reviewer 1 Report
No comments
Author Response
English language and style were checked.
Reviewer 2 Report
The authors largely did not provide the improvement requested.
The validation of MRAP2 antibody was not shown and while they state they validated it the data is not presented. MRAP2 KO or untransfected cells should be used as control.
The cross linking experiment was not improved. again they claim that they have run additional experiments but do not show the data. An MRAP2 blot is also required to confirm that the band indicated by the arrow contains MRAP2.
The justification for using PKR1 KO mice is not sound. The fact that PKR1 KO are obese and insulin resistant is controversial since other reports show no phenotype. if the authors claim that those mice are obese and insulin resistant, the data should be shown (body weight and GTT). Other models of obesity and insulin resistant that are far better established than PKR1 KO could be used. regardless, it is not clear how this addresses anything relevant to their study on PKR2.
Author Response
The authors largely did not provide the improvement requested.
- The validation of MRAP2 antibody was not shown and while they state they validated it the data is not presented. MRAP2 KO or untransfected cells should be used as control.
The anti-MRAP2 antibody was used only to characterize the protein expressed in yeast and to evaluate the protein expression in mice tissues: adipocytes of wt and PKR1 knock-out mice and hypothalamus explants. The MRAP2 antibody validation has been done in mice tissues that do not express MRAP2 such as skin and colon. (Masato Asai, et al., “Loss of Function of the Melanocortin 2 Receptor Accessory Protein 2 Is Associated with Mammalian Obesity”. Science. 2013 Jul 19; 341(6143): 275–278. doi: 10.1126/science.1233000). We attach the image of the western blotting apologizing for not doing it before (Fig. S3).
- The cross linking experiment was not improved. Again they claim that they have run additional experiments but do not show the data. An MRAP2 blot is also required to confirm that the band indicated by the arrow contains MRAP2.
We attach a blotting (Fig. S4) obtained with antibodies against PKR2 to support the results presented in the blotting with anti-MRAP2 antibodies shown in figure 3. This confirm that the band indicated by the arrow contains the complex MRAP2-PKR2.
- The justification for using PKR1 KO mice is not sound. The fact that PKR1 KO are obese and insulin resistant is controversial since other reports show no phenotype. If the authors claim that those mice are obese and insulin resistant, the data should be shown (body weight and GTT). Other models of obesity and insulin resistant that are far better established than PKR1 KO could be used. Regardless, it is not clear how this addresses anything relevant to their study on PKR2.
We examined the PKR1 knock out mice because our aim was to see how MRAP2 expression was affected by the prokineticin system. In our enclosure we have the colony of PKR1 knock out mice and they allowed us to obtain very interesting data regarding the role of the prokineticin system in obesity. (Maftei D et al., “The balance of concentration between Prokineticin 2β and Prokineticin 2 modulates the food intake by STAT3 signaling”. BBA Advance, 2021). The PKR1 knock out mice were extensively characterized in our laboratory and the data obtained match with those obtained by the group of Nebigil (Szatkowski C. et al., Plos one 2013).
The body weight of 40 weeks age PKR1 KO mice results 40 % higher respect that observed in WT mice.
In insulin tolerance test, PKR1 KO mice are less able to handle glucose loading. PKR1 KO mice display impaired glucose tolerance (GTT): after glucose administration, higher blood glucose concentration are recorded respect to wild type mice.
Round 3
Reviewer 2 Report
Authors have largely addressed my concerns